# Analysis of the Photogrammetric Use of 360-Degree Cameras in Complex Heritage-Related Scenes: Case of the Necropolis of Qubbet el-Hawa (Aswan Egypt)

**DOI:** 10.3390/s24072268

**Published:** 2024-04-02

**Authors:** José Luis Pérez-García, José Miguel Gómez-López, Antonio Tomás Mozas-Calvache, Jorge Delgado-García

**Affiliations:** Departamento de Ingeniería Cartográfica, Geodésica y Fotogrametría, Universidad de Jaén, 23071 Jaen, Spain; jlperez@ujaen.es (J.L.P.-G.); jglopez@ujaen.es (J.M.G.-L.); jdelgado@ujaen.es (J.D.-G.)

**Keywords:** fisheye lens, spherical image, panoramic image

## Abstract

This study shows the results of the analysis of the photogrammetric use of 360-degree cameras in complex heritage-related scenes. The goal is to take advantage of the large field of view provided by these sensors and reduce the number of images used to cover the entire scene compared to those needed using conventional cameras. We also try to minimize problems derived from camera geometry and lens characteristics. In this regard, we used a multi-sensor camera composed of six fisheye lenses, applying photogrammetric procedures to several funerary structures. The methodology includes the analysis of several types of spherical images obtained using different stitching techniques and the comparison of the results of image orientation processes considering these images and the original fisheye images. Subsequently, we analyze the possible use of the fisheye images to model complex scenes by reducing the use of ground control points, thus minimizing the need to apply surveying techniques to determine their coordinates. In this regard, we applied distance constraints based on a previous extrinsic calibration of the camera, obtaining results similar to those obtained using a traditional schema based on points. The results have allowed us to determine the advantages and disadvantages of each type of image and configuration, providing several recommendations regarding their use in complex scenes.

## 1. Introduction

The methods, techniques and sensors used to graphically document heritage have undergone a great development during the last decades, allowing us to obtain reliable models of reality in order to represent sites and artifacts. This documentation is mainly focused on conservation tasks but also other purposes such as virtual museums, dissemination, etc. There are several aspects that have contributed to this evolution, such as the development of new geomatic techniques, new sensors and data acquisition methodologies, the application of new algorithms, such as the Structure from Motion (SfM) [1,2,3] and the dense MultiView Stereo 3D reconstruction (MVS) [4,5,6,7], their implementation in several commercial applications, such as Agisoft Metashape and Pix4DMapper [8,9], the increase in the computing capabilities, etc. Regarding current geomatic techniques, we can highlight the generalized use of Light Detection and Ranging (LiDAR), particularly using Terrestrial Laser Scanning (TLS) in the case of indoor scenes, and non-metric conventional cameras (such as pinhole or perspective cameras) to develop photogrammetric surveys based on close range photogrammetry (CRP). This latter case has allowed a significant cost reduction compared to the use of metric cameras. In addition, the development of new platforms such as the Unmanned Aerial Vehicles (UAVs) [10,11,12] have made it possible to elevate these sensors to higher points of view, facilitating the capture of the scene. Some authors, such as Remondino [13] and Hassani et al. [14], have analyzed these techniques extensively, showing their advantages and disadvantages, although there are many applications that propose their combined use to take advantage of their integration [15,16,17,18,19,20,21,22].

Despite the current development of these techniques, widely applied in several disciplines, in recent years there have been some innovations that are focused on improving capture efficiency. For example, these new systems include mobile mapping systems (MMS) based on simultaneous localization and mapping (SLAM), which allow real-time capture, supporting data acquisition in inertial systems and/or GNSS (outdoor applications) and providing the trajectory and orientation of the system at any time. However, in some situations, the use of these new techniques is not possible due to their higher current cost and other obstacles.

In heritage, the presence of complex scenes is common, especially in interior spaces. These scenes can be defined as those whose geometrical characteristics, location and accessibility make a simple data acquisition difficult or impossible (e.g., narrow spaces with little distance between sensor and object). In these cases, the efficiency of data acquisition is one of the main aspects to be considered due to the difficulties in obtaining a complete coverage of the object by geomatic techniques that involve a large number of photographs (in the case of pinhole cameras) or scanning stations (in the case of using TLS). More specifically, in the case of photogrammetry, the number of photographs required to cover a complex scene can be reduced by using lenses that provide a larger Field of View (FoV), such as those using wide angle lenses [23,24,25,26,27], fisheye lenses [28,29,30,31,32,33,34,35,36,37,38] and 360-degree cameras [34,39,40,41,42,43,44,45,46]. In this context, spherical photogrammetry [47] has undergone a great development recently using both fisheye images (FEI) and spherical images (SI), which are also known as panoramic images [47,48,49,50,51,52].

Spherical images are obtained through the superposition of images obtained from a specific point of view in several directions until covering a larger FoV (some authors suggest values higher than 160 degrees horizontally to be considered as a panoramic image). To obtain them, we can use a conventional camera, rotating it around the lens axis in a defined direction obtaining several images that are merged and projected on a sphere, in most cases using the equirectangular projection, by means of geometrical and stitching procedures. The alternative and most efficient method to obtain these spherical images is to use a 360-degree camera, which can project the image obtained with a fisheye lens or by fusing and projecting several images from several fisheye lenses. Considering the latter case, several authors classified these cameras into three types: dioptric, catadioptric and polydioptric cameras [52,53,54]. The most commonly used in heritage applications are polydioptric cameras, which contain several fisheye lenses to capture scenes with a 360-degree horizontal FoV. Depending on the number of lenses (minimum 2), the system can achieve a greater overlapping of images, facilitating and improving the stitching procedure by using central zones where lens distortion is usually lower. In addition, the overlapping allows obtaining other relevant information such as depth maps because each point of the scene is captured from several points of view. At the moment, spherical photogrammetry has been applied in several studies, achieving accuracies of various centimeters [44,46,54,55,56,57,58]. Recently, the use of these sensors has been increasing due to their inclusion in MMS based on visual SLAM [59,60,61].

Regarding stitching techniques to obtain spherical images in a 360-degree camera, there are several classifications [62,63,64,65] taking into account some aspects related to the fusion and projection of images. In general, there are direct pixel-based methods and, most commonly, feature detection methods, which are based on algorithms such as the Scale-Invariant Feature Transform (SIFT) [3], and the Speeded Up Robust Features (SURF) [66], among others. In the case of complex scenes, we must consider several issues. Thus, differences in depth of the objects with respect to the camera may cause undesirable effects (blurring, ghosting, etc.) in the results of the stitching procedure due to different parallax, which is greater at shorter distances. To solve this, some authors [67,68] suggested the use of depth maps to improve the stitching results, which can be obtained thanks to the overlapping provided by polydioptric cameras.

A fundamental aspect in photogrammetry is the geometric calibration of the sensor, in this case, specifically related to fisheye sensors [69]. This calibration is basic to improving the geometric quality of the photogrammetric products through the determination of the intrinsic parameters of each sensor or interior orientation parameters (IOPs) (focal length, principal point and distortion parameters) and extrinsic or relative orientation parameters (ROPs) (rotation matrix and translation vectors between sensors). The determination of the extrinsic parameters is an element that contributes positively to the simplification of the final image orientation processes, considering that the mounting of the different cameras is sufficiently stable or robust, as in the case of 360-degree multi-cameras. Some studies [70,71,72] used these parameters to define constraints between those sensors, improving geometrical results.

In general, calibration methods are based on points and lines, patterns and self-calibration procedures. There are some studies related to calibration applied to fisheye lenses [73,74,75,76,77,78,79,80] and 360-degree cameras [51,69,70,81,82,83], which have improved geometric results. Other studies showed comparisons between several types of cameras (perspective, fisheye, etc.) [34,84]. In any case, intrinsic calibration allows an improvement in accuracy. On the other hand, extrinsic calibration allows obtaining the relationships between the different elements that make up the sensor, simplifying the final orientation procedure of the images as a whole while allowing more robust orientations to be obtained [69,81,83].

The current development of 360-degree cameras (improved sensor spatial resolution) allows their use for heritage documentation by facilitating data acquisition tasks thanks to the increased coverage (FoV) of the sensors. They can be used in complex scenes, minimizing the number of images needed to cover the entire scene. However, some studies have shown certain limitations related to the geometric accuracy when compared to other techniques based on TLS or conventional photogrammetry. In the case of indoor applications, there are other difficulties related to illumination conditions, mainly when the radiometric aspect is important, as is common in heritage studies. In these areas, the definition of a reference coordinate system (CRS) based on using surveying techniques to georeference data is also difficult, and in this sense, the reduction of these tasks is an important challenge to be faced. Another important aspect is related to the selection of the type of image (fisheye and spherical) and the stitching technique (spherical images) to be used. Although some of the studies developed so far have analyzed some of these aspects, they have addressed them partially (e.g., analyzing the stitching techniques) and not always oriented to complex heritage-related scenes, where there are major complications, as described previously. In this study, we analyze the different aspects of the use of 360-degree cameras and spherical photogrammetry [47] in complex scenes from multiple points of view: type of image, geometrical quality, minimization of GCPs, etc. As an example, the analysis includes the use of fisheye images directly and spherical images generated from them. The goal is to determine the aspects to be considered in order to improve the results, mainly in the photogrammetric orientation processes, taking into account the improvement of the efficiency in the data capture and the minimization of auxiliary work.

The objectives of this study are related to the analysis of the application of 360-degree cameras and spherical photogrammetry to heritage sites in order to improve the quality of products and the efficiency in data acquisition, reducing the number of GCPs needed for orientation processes. The analysis mainly focuses on cases of complex scenes, poorly illuminated and with short and variable distances between the sensor and the object. In this sense, the methodology to be employed should analyze which type of image (fisheye or spherical) is more convenient in these cases to improve geometric results and reduce surveying work.

The manuscript is divided into three sections. The first section comprises the presentation of the method developed to analyze these images and the materials used to apply it to some specific complex scenes of funerary structures in Egypt. The second section describes the main results obtained after the orientation of the spherical images considering several stitching techniques and fisheye images. In addition, this section also includes the results after using fisheye images considering certain constraints determined between all sensors thanks to the extrinsic calibration carried out. The final section includes the main conclusions and the proposals for future work.

## 2. Materials and Methods

The methodology proposed in this study (Figure 1) analyzes the accuracy results achieved after the image orientation processing by considering several cases related to the use of fisheye images (FEI) and spherical images (SI) obtained from the same 360-degree multi-camera. Thus, this comparison is developed using images obtained from the same acquisition considering several complex scenes related to heritage.

Prior to the development of the photogrammetric process using fisheye and spherical images, we must consider an important aspect related to the characteristics of these images, which is considered as an initial hypothesis in this study. The fisheye images are obtained directly from the sensor, while the spherical images are synthetic, obtained by transformation (or re-projection) and fusion. This implies that the spherical images are not associated with any sensor with a defined internal geometry. In the methodology proposed in this study, we use three stitching techniques to obtain these images, two based only on feature-based techniques (SI-FS and SI-HQS in Figure 1) and one that additionally considers depth maps (SI-DM in Figure 1). As discussed in the previous section, feature-based techniques can show problems due to parallax, different distances between objects and the sensor, etc., which cause geometrical errors in the final stitched image [67]. However, the spherical images based on depth maps show a more realistic representation because the true depth of the object with respect to the sensor is known and considered at all projected points [67,68]. This aspect contrasts to other techniques that use a single distance between the sensor and the object, not considering depth maps and incorporating this simplification into the image processing. Therefore, these assumptions are considered in the analysis proposed in this study.

The procedure (Figure 1) starts with the acquisition of fisheye images using a 360-degree multi-camera. The device must be located at different locations to guarantee full coverage of the scene. At each position, we obtained a set of fisheye images equal to the number of sensors contained in the multi-camera. These images have full overlapping that facilitates stitching procedures. After that, three spherical images were obtained from the set of fisheye images considering three stitching procedures (SI-FS, SI-HQS and SI-DMS). As an example, a comparison of tie points determined by two stitching methods (SI-HQS and SI-DMS) was performed using the distances determined between them in order to show their geometrical discrepancies. After that, the photogrammetric orientation of these blocks of images and the block that contains the fisheye images (FEI) was carried out by means of several well-distributed ground control points (GCPs), materialized on the scene by targets, whose coordinates were obtained from a surveying network using a total station. The results of these four block orientations were compared using the calculated Root Mean Squared Error (RMSE) of a set of check points (CPs).

Next, we compare the results of fisheye image orientation both using and not using GCPs. When GCPs are not used, we introduce sensor extrinsic calibration parameters (relative orientation) to provide several constraints to the system. Specifically, in this case, we set the block scale using scale bars (SBs). The accuracy of the orientation (FEI-SBs) is calculated by analyzing the residuals obtained in a 3D rigid transformation. This transformation is based on several GCPs defined in the images (targets). In addition, the two 3D meshes of each scene obtained from the point clouds determined from both photogrammetric blocks (FEI-GCPs and FEI-SBs) are aligned in the same CRS and compared, showing the discrepancies (distances) from mesh to mesh. The methodology proposed in this study is applied using a specific 360-degree multi-camera in a specific heritage site that is described below.

### 2.1. Materials: 360-Degree Camera

In this study, we used a Kandao Obsidian Go 360-degree camera (Figure 2a) due to the possibility of using depth maps to develop the stitching procedure. In fact, this is one of the few 360-degree cameras that can generate depth maps [85]. The camera has a focal length of 6.8 mm. The fisheye image obtained has a resolution of 4608 × 3456 pixels. The camera is composed of six fisheye lenses with a horizontal coverage of about 220 degrees each (Figure 2b). Their optical axes are distributed with an angle of 60 degrees. The horizontal FoV allows obtaining full coverage (360 degrees) using all sensors with a large overlap between them. In this sense, this sensors configuration provides two adjacent images with an approximate overlap of 160 degrees. Thus, 70% of the scene is covered by four sensors, and the remaining 30% is covered by three lenses. Consequently, the use of the six images makes it possible to obtain a depth map of the scene thanks to these large overlaps. The software used to obtain the spherical images using the six fisheye images captured in each acquisition is Kandao Studio v2.7 (Figure 2c). This application allows obtaining spherical images (SI-FS, SI-HQS and SI-DMS) (resolution of 7680 × 3840 pixels) using three stitching methods: Fast Mode, High Quality Mode and Depth Mode, and developing an internal pre-calibration process.

### 2.2. Materials: Images

As discussed in previous sections, we used two types of images: fisheye and spherical images, the latter being obtained by three different methods. Considering a camera capture at a specific location, Figure 3 shows some examples of the six fisheye images acquired and the three spherical images determined using the three stitching options available in the Kandao Studio software v2.7 (Fast Stitching -SI-FS-; High Quality Stitching -SI-HQS-; and, Based on Depth Maps -SI-DMS-). In the case of fisheye images, the intrinsic parameters (focal length, principal point and distortion) and extrinsic parameters (rotation matrix and translation between sensors) are calculated through calibration techniques using GCPs.

### 2.3. Materials: Scenes

The proposed methodology has been applied to a real study case composed of several complex scenes in four rock-cut funerary structures at the Necropolis of Qubbet el-Hawa (Assan, Egypt) (Figure 4a). This site is located in a hill situated on the west bank of the Nile River (Figure 4b), and it is composed of more than one hundred tombs of different dimensions (Figure 4c). The larger ones are distributed among several rooms, such as halls of pillars, corridors, vertical shafts and burial chambers, while the smallest are composed of a simple burial chamber (Figure 4d). Both cases include certain characteristics that complicate the application of photogrammetric techniques to document them. Thus, they contain narrow and reduced spaces to capture images and a complex geometry with pillars, niches, etc. (Figure 4d) that hinder the use of conventional photogrammetry. In this case, the use of spherical photogrammetry presents several advantages like reducing the number of images needed and other additional work such as GCP measurements. One of the objectives proposed for this research is to analyze which of the types of images considered (FEI or SI) are most suitable for this type of archaeological documentation work in complex areas.

In this study, we have selected four scenes to obtain more conclusive results. In this sense, we have considered some geometrical aspects, trying to include some specific structures such as vertical shafts, corridors and the presence of different chambers separated by narrow accesses. Table 1 shows the main characteristics of the scenes used in this study, including the total number of captured images, the average distance between the 360-degree camera and the object and the total number of GCPs and CPs used.

Figure 5 shows the geometry of these scenes and the locations (top view) of the camera, GCPs and CPs. These points were materialized using targets whose coordinates were calculated from a surveying network using a total station.

Considering that these burial structures are distributed throughout the inside of the hill, another important aspect to be considered is the illumination of the scenes in order to obtain images with a similar radiometric response. To solve this issue, we mounted an LED lamp connected to an external battery in the upper part of the camera (Figure 6), aiming to minimize both the zones where the illumination system appears in the images and the overexposed areas due to the proximity between the camera and the object.

## 3. Results and Discussion

### 3.1. Fisheye Images vs. Spherical Images

#### 3.1.1. Fisheye Sensor Intrinsic Calibration

The use of FEI requires the knowledge of the internal geometry of the sensors and consequently, the development of intrinsic sensor calibration. In this study we carried out the intrinsic calibration of the selected 360-degree camera using a defined 3D pattern composed of a set of GCPs, materialized by targets. The coordinates of these points were previously calculated using TLS, and more specifically with a Faro Focus X130 scanner. This pattern was captured by the 360-degree camera in five different positions (Figure 7).

The images captured with the 360-degree camera have been oriented using the GCP set using the Agisoft Metashape v2 software, obtaining the intrinsic parameters (focal length [f], position of the main point [cx and cy] and radial distortion parameters [K1 to K4]) that are shown in Table 2. These parameters are related to the Brown distortion model.

#### 3.1.2. Spherical Images

The comparison between the three stitching techniques is carried out using a set of keypoints obtained after applying feature point matching to the three blocks of images. The keypoints are compared by selecting those that are homologous between two images (tie points) obtained from different stitching procedures. Therefore, we avoid those keypoints that have no correspondence in the pair of images to be compared. After that, we obtained a displacement vector for each point and consequently the geometric discrepancies of the images of the same scene depending on the stitching technique used to generate the spherical images. Figure 8 shows an example of this comparison between spherical images obtained using and without using depth maps (Figure 8a,c). Figure 8b,d show detailed views of some displacement vectors calculated between homologous points located in the SI-HQS and the SI-DMS, respectively.

Figure 9 shows a frequency line chart of the distances obtained using more than 1200 tie points measured on both images (SI-HQS and SI-DMS). This analysis results in an average distance value of about 34.5 pixels and minimum and maximum values of 1.4 and 168.9 pixels, respectively.

#### 3.1.3. Orientation Using GCPs

The previous analyses show large geometrical differences between spherical images obtained by different stitching techniques. To analyze their behavior with respect to the original fisheye images, we carried out an experiment consisting of studying the results of the photogrammetric block orientation procedure considering the four types of images previously described (SI-FS, SI-HQS, SI-DMS and FEI cases) (Figure 3) in all scenes (Figure 5). Thus, we used GCPs to carry out the orientation of image blocks and CPs to analyze the results of this procedure. These sets of points are well-distributed throughout the scene (Figure 5). In the FEI case, the intrinsic parameters (Table 2) are readjusted during the orientation procedure. The analysis is performed using Agisoft Metashape v2 software. Figure 10 shows the RMSE values of the four cases and the average values of all scenes. The graph shows a clear reduction of RMSE values from those cases of spherical images that do not consider depth maps to SI-DMS and FEI in all scenes (Figure 10a). More specifically, the average RMSE values are reduced in the case of using SI-HQS instead of SI-FS (from 11.3 pixels to 7.2 pixels, which is about 36%) and more in the case of using SI-DMS (from 7.2 pixels to 1.3 pixels, about 81%). In addition, Figure 10b shows a detailed view with a reduction of RMSE values from SI-DMS to FEI (1.3 to 0.6 pixels, about 54%).

The results show a large difference in RMSE values depending on the image types. Thus, there is a great improvement in residuals when using SI-DMS and FEI, so it can be recommended not to use the first two types of stitching (SI-FS and SI-HQS) when the use of depth maps (SI-DMS) is available. Otherwise, the SI-HQS should be selected when spherical images are to be used. Considering the cases SI-DMS and FEI the reduction of the average RMSE between SI-DMS and FEI is less than one pixel. Nevertheless, the use of FEI involves the processing of blocks made up of a greater number of images with respect to SI-DMS. This implies a longer processing time in all photogrammetric processes. Therefore, both cases show advantages and disadvantages, which are discussed in the next section.

#### 3.1.4. Advantages and Disadvantages of Using Spherical and Fisheye Images

Currently, only a few 360-degree cameras allow spherical images to be obtained using depth maps due to the need for full overlapping between fisheye images, and as a consequence, the need for multiple sensors (more than two fisheye lenses). In fact, most cameras marketed so far have only two fisheye lenses, positioned opposite each other to cover 360 degrees but without full overlapping. Therefore, this configuration makes it impossible to obtain complete depth maps. We consider the use of SI-DMS limited to 360-degree multi-cameras with a larger number of fisheye sensors, which provide extensive overlapping between images. Other 360-degree cameras should use stitching techniques that do not consider depth maps.

The use of spherical images instead of the original fisheye images implies a large reduction in the number of images, although we eliminate the redundancy of information in each capture provided by the fisheye images. However, the processing time is reduced with respect to fisheye images where the number of images of the block is multiplied by the number of sensors. In addition, the use of spherical images does not require the knowledge of the internal geometry of the sensor (intrinsic calibration) because they are synthetic images obtained by projecting onto a sphere.

The use of FEI implies the need to know the intrinsic parameters of the sensor. The geometry of these cameras provides a certain redundancy of information in overlapping areas, improving the geometrical results. Despite this advantage, the use of FEI implies an increase in processing time. We have also detected greater difficulties in the orientation processes that are resolved by including manual tie points, but consequently increasing the time dedicated to this process.

Therefore, the selection of the type of image to use will consider the requirements of each project, taking into account accuracy and processing time, because the use of FEI allows obtaining more accurate results but involves more processing time.

Furthermore, from the improved accuracy in the orientation processes, the use of FEI adds another advantage over spherical images. This is the possibility of eliminating the use of GCPs or other external systems to scale the photogrammetric block. This is possible in the case of FEI through the use of intrinsic and extrinsic calibration of the camera, where the internal geometry and the relative position of all sensors is known. In this sense, the distance between all sensors can be calculated, and these values can be used to scale the scene without using external information. In this context, minimizing the application of surveying techniques to obtain the GCPs coordinates is an important advantage that is analyzed in the next section.

### 3.2. Using FEI without GCPs

#### 3.2.1. Extrinsic Calibration

The 360-degree camera contains several fisheye sensors mounted on a stable platform. The relative orientation between these sensors can be obtained by extrinsic calibration of the camera. Knowledge of the extrinsic parameters can facilitate the orientation processing because the distances between sensors can be used as system constraints, for example, in the case of Agisoft Metashape v2 software by including them as scale bars (FEI-SBs). In the case of the 360-degree camera used in this study, we used the same 3D pattern developed to calculate the intrinsic parameters (Figure 7). As results, Table 3 shows the average values of the calculated distances between all sensors and the standard deviation (STD) derived from this calculation.

#### 3.2.2. Orientation Using FEI without Using GCPs

The purpose of this analysis is to test the 360-degree camera orientation procedures without using GCPs. To verify the results obtained, the blocks have been oriented using two different schemes. Firstly, orientation was performed without considering GCP but incorporating the information derived from the distances previously calculated in the extrinsic calibration process, using the software option to incorporate scale bars (FEI-SBs). In this case, the intrinsic parameters (Table 2) are readjusted during the procedure, while the extrinsic parameters (Table 3) are used as constrains and are not recalculated, although they consider their quality (given by the STD in Table 3). Secondly, these orientations have been performed using a set of control points well distributed throughout the scene (see Section 3.1.1). Figure 11 shows an example of a block related to zone QH33SB (Figure 11a) and the scale bars included in two camera acquisitions (Figure 11b).

#### 3.2.3. Three-Dimensional Rigid Transformation

In order to verify the results of using scale bars as a constraint for orientation processing, we used the targets as CPs. For that purpose, the CPs were measured on the images (FEI-SBs) after orientation, and their coordinates were obtained in a local CRS. In order to make the comparison, both systems (global coordinates in GCP and local coordinates in the orientation that do not use GCP) were adjusted using a three-dimensional rigid transformation (six parameters including three translations and three rotations—omega, phi and kappa. The adjustment parameters obtained and, in a special way, the distances calculated at each of the points provide us with information of great interest when establishing the quality of the orientation. Figure 12 shows the results of the average, standard deviation, minimum and maximum values of the 3D distances of all scenes. In all cases, the average value is less than 0.02 m. The maximum values of QH33SP and QH35P are approximately 0.03 m, while QH23 is between 0.015 and 0.02 m. The maximum value of QH33SB is about 0.01 m. Therefore, these results indicate a certain geometrical similarity between these points in both CRSs and, as a consequence, in the accuracy achieved using the two orientation techniques (GCPs or scale bars).

#### 3.2.4. Modeling and Comparison of 3D Meshes

The previous analysis is based on specific points (targets), which are affected by errors of the surveying technique developed. In order to analyze a large number of points and minimize these errors, we compared the 3D meshes obtained using FEI-CGPs and FEI-SBs in all scenes. The 3D mesh obtained using scale bars was referenced to the CRS of the project using a registration divided into several stages and performed using Maptek Point Studio v2022 software. Firstly, we performed a translation and three rotations (Figure 13a,b) in order to place the FEI-SBs mesh approximately aligned with the FEI-GCPs mesh. Subsequently, we developed an automatic adjustment of the FEI-SBs mesh to the other to ensure that both meshes had the same CRS (Figure 13c). Finally, the distances between both meshes were calculated (Figure 13d).

Figure 14 shows the frequency line charts of distances between both meshes and the average distances and standard deviations obtained in all scenes. The differences are lower than 1 cm in most cases, with QH35P showing the highest discrepancies. Therefore, the results have demonstrated the possible use of FEI-SBs, avoiding or minimizing the use of GCPs, which is very interesting in order to improve the efficiency of field work.

### 3.3. Processing Times

This section discusses an example of processing times related to each procedure, taking into account all scenes used in this study. Three procedures were considered: stitching, relative orientation and marker projections (measurement of GCPs in all images). Table 4 shows the mean values for the four scenes, including the average number of images, the average number of GCPs and the average times (in seconds) spent for these procedures. The values were calculated using a computer with an i7-13700H CPU at 2.4 GHz, Geforce RTX 4070 GPU and 32 GB RAM. The total time column shows lower values for SI cases with respect to FEI cases. More specifically, the time differences between SI cases are not very significant and much less when comparing SI-HQS and SI-DMS. Considering FEI cases, the use of GCPs involves a large increase in time with respect to not using GCPs (almost three times longer). This case (FEI-SBs) spent about 25% more time than SI cases. However, this increase is largely compensated by the reduction of surveying fieldwork.

## 4. Conclusions

In this study, we have analyzed the use of 360-degree cameras, considering several cases and aspects related to spherical photogrammetry in complex scenes and more specifically in funerary structures characterized by reduced dimensions and poor illumination conditions. To carry out this study, we have focused on a specific polydioptric camera (Kamdao Obsidian Go) that allowed us to obtain spherical images using depth maps. The analysis has included all possible types of images related to this camera, the original fisheye images and three types of spherical images obtained using three different stitching techniques. The geometric errors obtained in the analyses show that the use of these types of images is suitable for most of the studies related to heritage documentation. Their use represents a great advantage with respect to the use of conventional photogrammetric techniques based on pinhole or perspective cameras due to the improvement of the capture tasks by reducing the number of images needed to cover the scene completely. In our opinion, the use of spherical photogrammetry is more suitable in cases of complex spaces due to the reduction of acquisition and processing time and consequently costs (Table 4). The selection of fisheye images or spherical images will consider the requirements of each project, taking into account the advantages and disadvantages evidenced in this study, which are summarized in Table 5 and described below:Spherical images: The stitching technique selected will largely condition the geometric quality of these images and consequently the results. In this sense, we suggest the use of stitching techniques based on depth maps because this study has demonstrated a clear improvement in the results with respect to the others. This option is limited to sets of fisheye images with full overlaps, which are obtained with a larger number of sensors, such as the one selected in this study. In this regard, we recommend the use of 360-degree multi-cameras composed of more than two fisheye lenses to obtain full overlaps. Although the use of the original fisheye images has shown better results, the analysis of spherical images based on depth maps has shown sufficient accuracy for most heritage documentation studies, showing the advantage over fisheye images of a significant reduction in the number of images, and consequently orientation processing time (Table 4). However, these images require a stitching processing time. In addition, their orientation requires GCPs, which is a problem in complex scenes. In our opinion, spherical images can be used in blocks composed of a large number of images, where the distribution and measurement of GCPs is not a significant problem. In any case, in this context, the use of spherical images will be subordinated to the application of stitching techniques based on depth maps.Fisheye images: The results have shown a higher geometric quality in the orientation processes, although the larger number of images will mean a longer processing time (Table 4). We have also detected greater difficulties in orientation processes when we do not use constraints between sensors. In some cases, we have to include tie points manually to complete the relative orientation. On the other hand, the results in the scenes studied have shown that the use of the extrinsic parameters to determine constraints as a function of the distance between sensors (scale bars) facilitates the relative orientation processes and reduces the use of GCPs to those necessary for the block georeferenced using a 3D rigid transformation (minimum 3 points). This has been confirmed by the transformation residuals and by the comparison of 3D meshes obtained in all scenes. In summary, the reduction of field work related to surveying techniques and consequently in costs is evident. In our opinion, the use of fisheye images will be recommended in complex scenes similar to those studied in this case, using a previous calibration that allows defining distance constraints to facilitate orientation and scaling processes.

Future work will focus on adding new complex indoor and outdoor scenes of higher dimensions in order to analyze disorientation problems caused by drift error in the case of using FEI-SBs. We will also analyze radiometric aspects when using this type of image in complex indoor scenes. This aspect is very important in the cases of using these images to obtain realistic textures for modeling. We also suggest the analysis of images extracted from video to improve capture efficiency, and more specifically to consider these acquisition techniques in complex indoor trajectories.

## Figures and Tables

**Figure 1 sensors-24-02268-f001:**
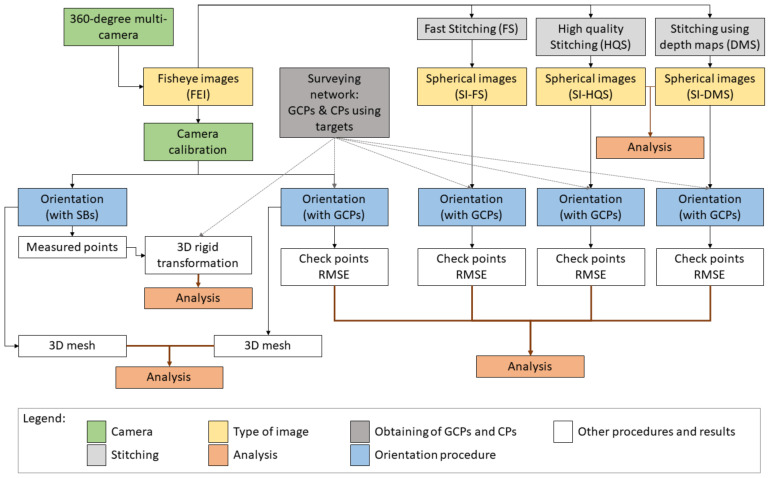
Methodology proposed in this study.

**Figure 2 sensors-24-02268-f002:**
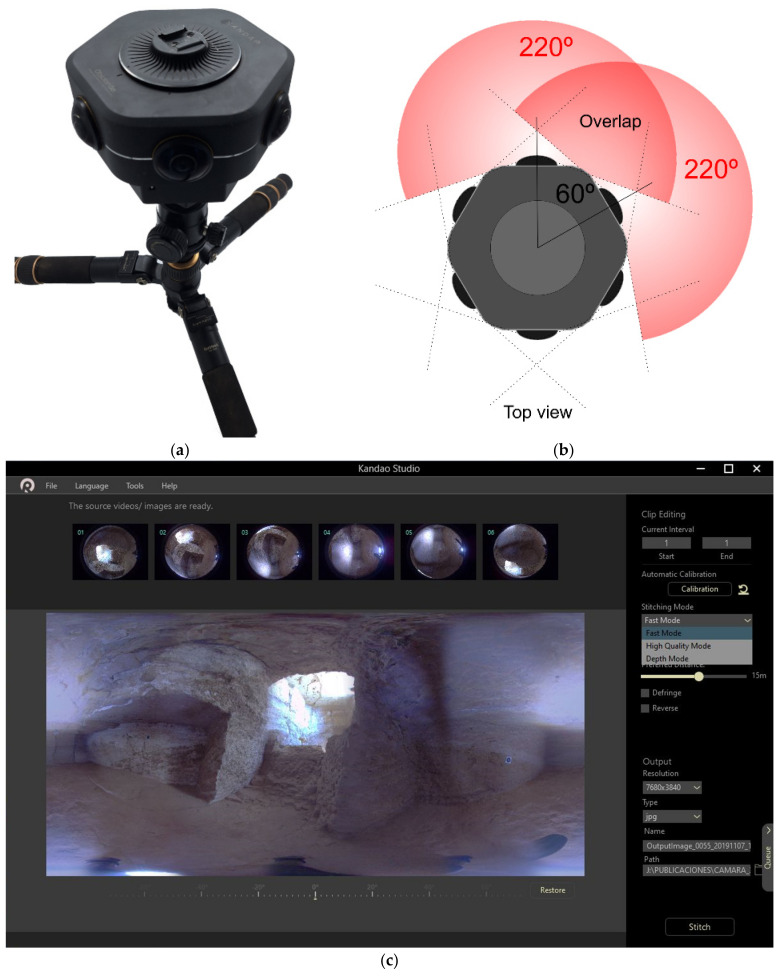
The 360-degree camera used in this study: (**a**) general view of the Kandao Obsidian Go camera; (**b**) top view scheme; (**c**) view of stitching menu of Kandao Studio v2.7.

**Figure 3 sensors-24-02268-f003:**
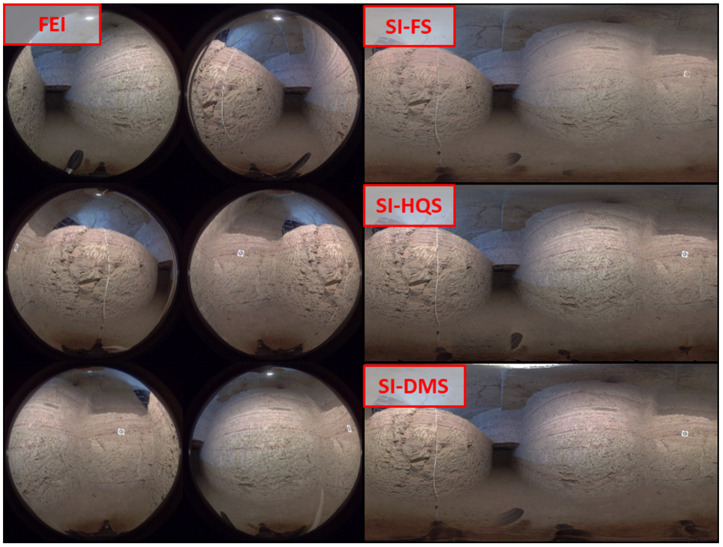
Types of images used in this study: Fisheye images (FEI), spherical images by fast stitching (SI-FS), spherical images by high-quality stitching (SI-HQS) and spherical images using depth map stitching (SI-DMS).

**Figure 4 sensors-24-02268-f004:**
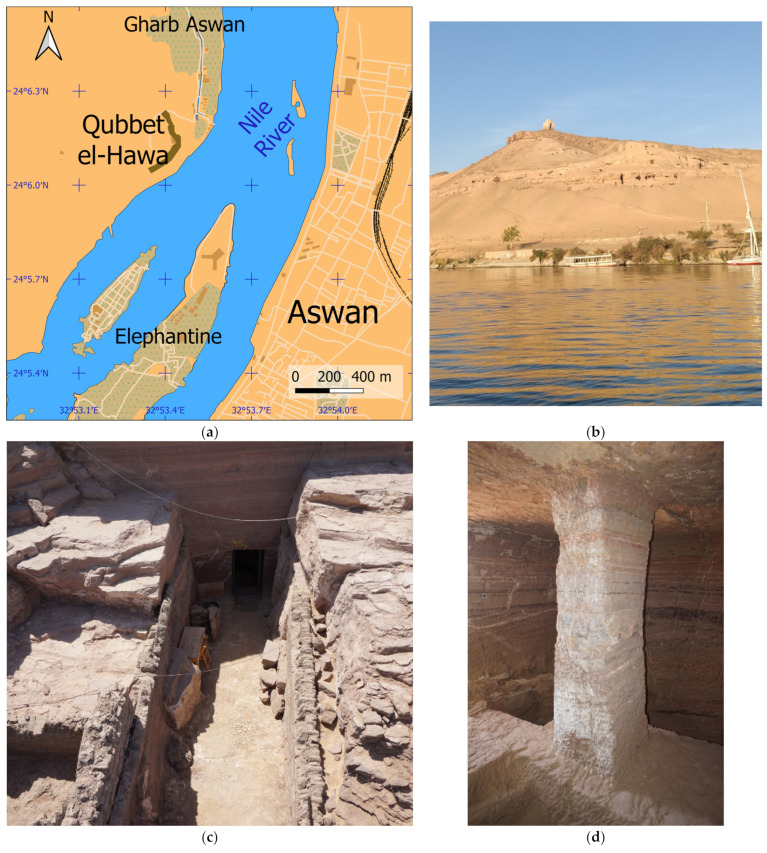
The Necropolis of Qubbet el-Hawa (Aswan, Egypt): (**a**) Location; (**b**) general view of the hill; (**c**) access courtyard of a burial structure; (**d**) burial chamber.

**Figure 5 sensors-24-02268-f005:**
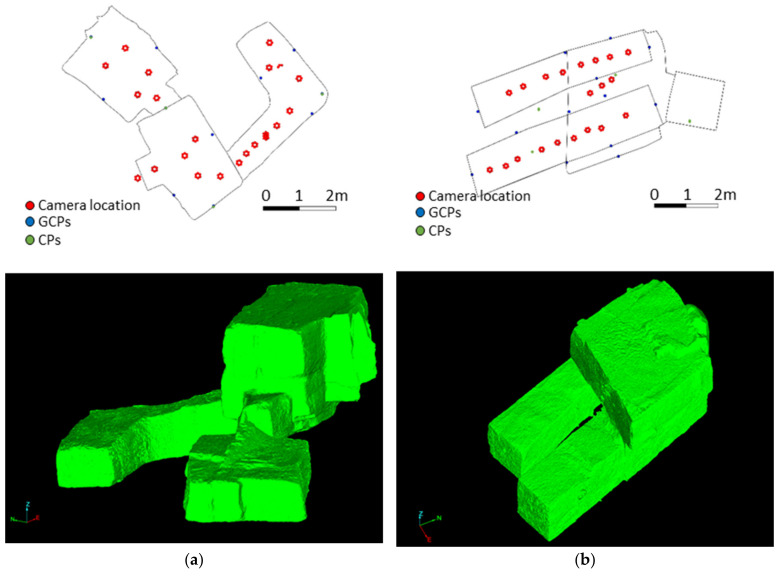
Scenes used in this study: (**a**) QH23; (**b**) QH33SB; (**c**) QH33SP; (**d**) QH35P.

**Figure 6 sensors-24-02268-f006:**
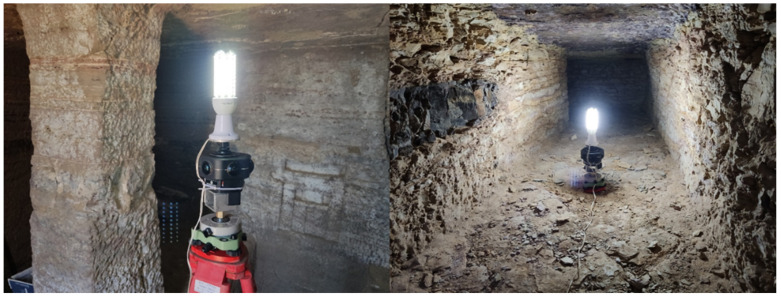
Examples of the illumination system mounted on the camera.

**Figure 7 sensors-24-02268-f007:**
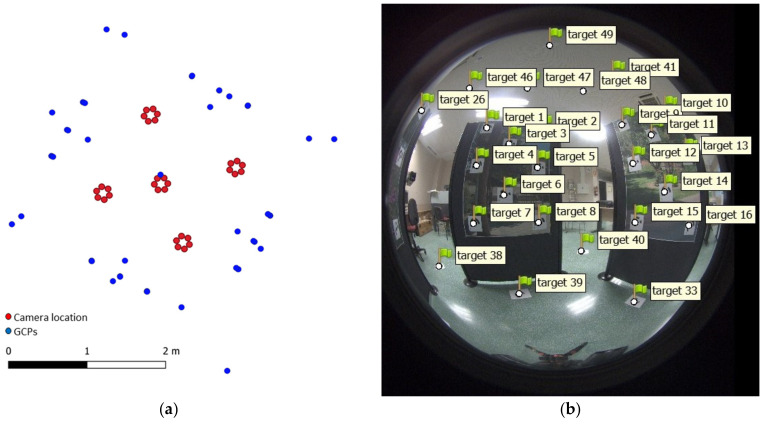
Calibration pattern. (**a**) Top view of the distribution of GCPs, camera positions and fisheye sensors. (**b**) Distribution of GCPs among one fisheye image captured from one position.

**Figure 8 sensors-24-02268-f008:**
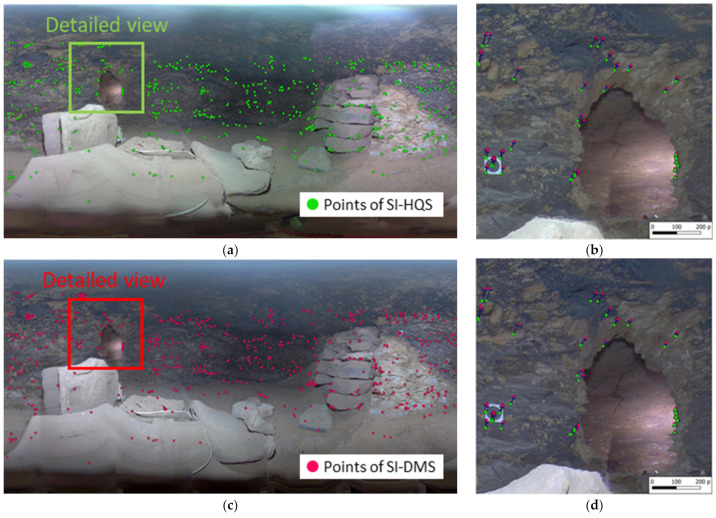
Displacement vectors obtained between homologous points extracted from SI-HQS and SI-DMS: (**a**) SI-HQS; (**b**) detailed view of SI-HQS; (**c**) SI-DMS; (**d**) detailed view of SI-DMS.

**Figure 9 sensors-24-02268-f009:**
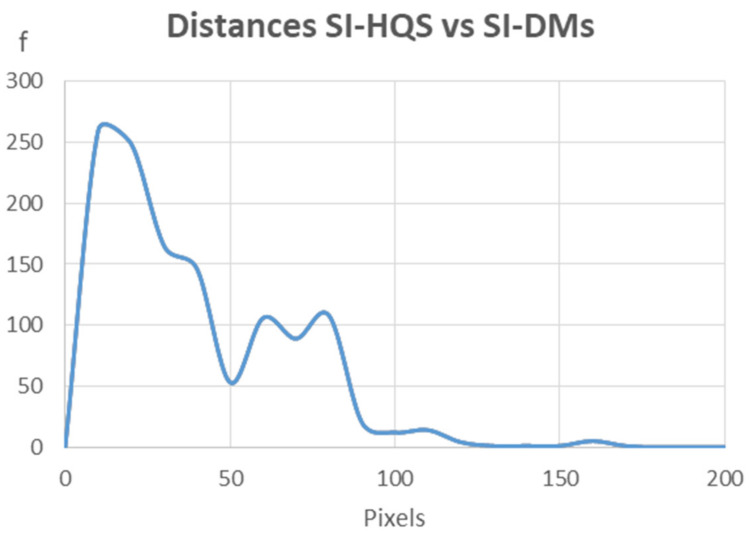
Histogram of distances between SI-HQS and SI-DMS.

**Figure 10 sensors-24-02268-f010:**
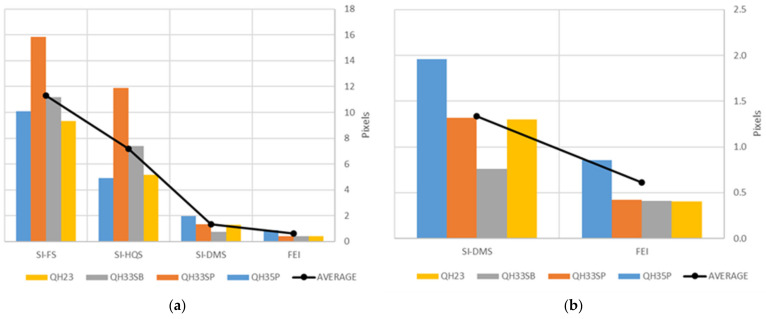
RMSE at the CPs for the different types of images and study areas considered: (**a**) All types of images; (**b**) SI-DMS and FEI.

**Figure 11 sensors-24-02268-f011:**
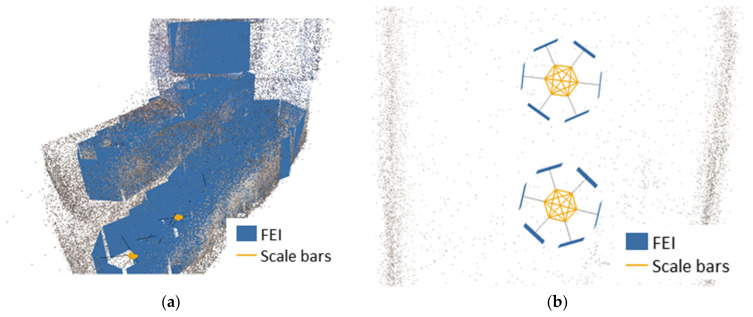
Photogrammetric orientation procedure using scale bars in Agisoft Metashape v2 software: (**a**) photogrammetric block; (**b**) scale bars in two captures.

**Figure 12 sensors-24-02268-f012:**
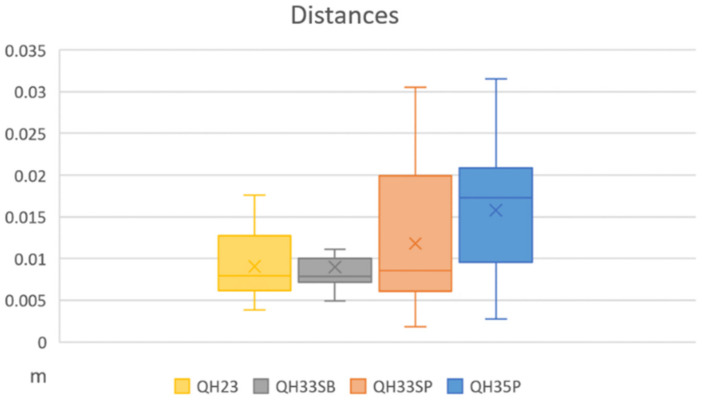
Distances obtained after transformation.

**Figure 13 sensors-24-02268-f013:**
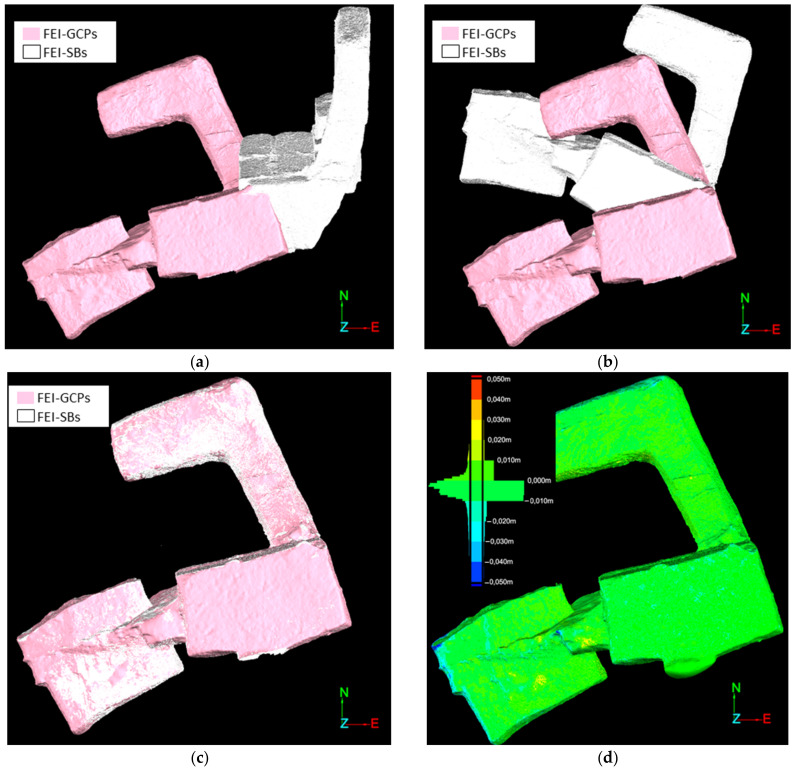
Example of meshes obtained using FEI-GCPs and FEI-SBs: (**a**) translation; (**b**) rotations; (**c**) automatic adjustment; (**d**) distances between meshes.

**Figure 14 sensors-24-02268-f014:**
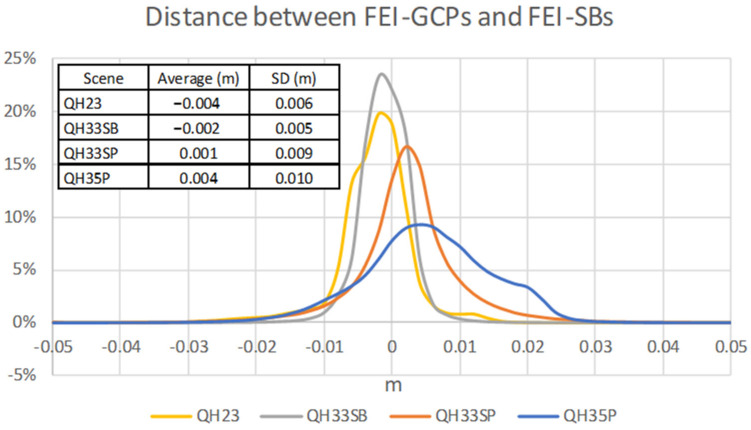
Histograms of distances between meshes.

**Table 1 sensors-24-02268-t001:** Characteristics of the scenes used in this study.

Scene	Captures	Average Distance (m)	GCPs	CPs
QH23	19	1.9	14	4
QH33SB	20	3.4	14	5
QH33SP	14	1.6	9	4
QH35P	28	2.1	19	4

**Table 2 sensors-24-02268-t002:** Intrinsic parameters.

Sensor	f (Pixels)	cx (Pixels)	cy (Pixels)	K1	K2	K3	K4
1	1110.667	23.296	19.106	−5.9095 × 10^−2^	−1.6487 × 10^−3^	1.2992 × 10^−4^	−8.3588 × 10^−7^
2	1112.981	−25.230	4.259	−6.0611 × 10^−2^	−6.6440 × 10^−4^	−3.1994 × 10^−4^	6.4145 × 10^−5^
3	1106.498	−25.919	34.177	−5.8917 × 10^−2^	−2.0134 × 10^−3^	1.3292 × 10^−4^	1.7086 × 10^−5^
4	1106.968	−26.215	−39.301	−5.9852 × 10^−2^	−6.9513 × 10^−4^	−4.8518 × 10^−4^	1.1370 × 10^−4^
5	1118.038	−57.224	0.222	−6.0680 × 10^−2^	−1.3719 × 10^−3^	7.0947 × 10^−5^	7.0671 × 10^−7^
6	1110.727	24.278	−60.529	−5.9773 × 10^−2^	−2.6219 × 10^−3^	6.0588 × 10^−4^	−7.5270 × 10^−5^

**Table 3 sensors-24-02268-t003:** Average distances between sensors and standard deviation.

	Distances (m)					STD (m)				
Sensor	2	3	4	5	6	2	3	4	5	6
1	0.0655	0.1132	0.1307	0.1134	0.0655	0.0002	0.0002	0.0002	0.0003	0.0003
2		0.0651	0.1131	0.1308	0.1133		0.0002	0.0002	0.0005	0.0001
3			0.0655	0.1134	0.1307			0.0002	0.0003	0.0003
4				0.0653	0.1130				0.0003	0.0001
5					0.0654					0.0002

**Table 4 sensors-24-02268-t004:** Summary of processing times.

Cases	Number of Images	Number of GCPs	Stitching Time (s)	Orientation Time (s)	Marker Projections Time (s)	Total (s)
SI-FS	22	15	76	63	568	707
SI-HQS	22	15	137	63	568	768
SI-DMS	22	15	143	63	568	774
FEI-GCPs	134	15	0	1084	1980	3064
FEI-SBs	134	0	0	1084	0	1084

**Table 5 sensors-24-02268-t005:** Summary of the advantages and disadvantages of using each configuration.

Stage	SI	FEI-GCPs	FEI-SBs
Pre-processing (obtaining images)	Stitching	No	No
Calibration	No	Intrinsic	Complete
Orientation	Without redundancy	Higher redundancy. Problems to complete the relative orientation	Higher redundancy
GCPs (photogrammetric)	Yes	Yes	No
Transformation from a local CRS	No	No	Yes

## Data Availability

Data are contained within the article.

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
