# Peer review of "Analysis of the Photogrammetric Use of 360-Degree Cameras in Complex Heritage-Related Scenes: Case of the Necropolis of Qubbet el-Hawa (Aswan Egypt)"

_sensors, 2024, doi:10.3390/s24072268_

Round 1

Reviewer 1 Report

Comments and Suggestions for Authors

Taking the necropolis of Qubbet el-Hawa as an example, the authors analyze the results of photogrammetric measurements applied to complex heritage-related scenes using a 360-degree camera. The analysis compares the effects of using different types of images, various stitching methods, and the utilization of ground control points on the measurement results. The article has a clear and logical structure overall; however, there are the following issues:

1、The accuracy and logic of the article are insufficient. The expressions in some sentences are not clear enough. There is an issue with sentences being overly lengthy, which makes it difficult to grasp the main points, and there are problems with the conjunctions used between some sentences.

2、The references cited in the article are not up-to-date.

3、The research subject of the article is complex heritage-related scene. How does the author define "complex"? Are there any quantitative standards?

4、The article selects four scenes for experimentation. What is the basis for selecting these four scenes? What are their respective characteristics, and what are the difficulties in conducting photogrammetric measurements? Do they have representativeness?

5、Table 2 lacks an introduction to the column headers.

6、The description of the experimental results is not accurate enough. In lines 418-419 of the article, it is mentioned that "in all scenes, the maximum values are about 0.03m." However, as shown in Figure 12, only the last two scenes are around 0.03m.

Comments on the Quality of English Language

The quality of English needs to be improved. Please revise the article throughout.

Author Response

Authors: Thank you for your suggestions and comments. We have rewritten the paper following your suggestions.

Reviewer 1: 1、The accuracy and logic of the article are insufficient. The expressions in some sentences are not clear enough. There is an issue with sentences being overly lengthy, which makes it difficult to grasp the main points, and there are problems with the conjunctions used between some sentences.

Authors: We have simplified and rewritten some sentences to improve the clarity of the article. Thank you.

Reviewer 1: 2、The references cited in the article are not up-to-date.

Authors: We have reviewed the list of references and most of them are from the last 10 years. The remaining references are mainly related to algorithms and methods defined longer ago and cannot be updated. In addition, we have reviewed more studies that are recent, but we consider that the current list fits better with the text. Please could you suggest which references should been changed and/or added? Thank you.

Reviewer 1: 3、The research subject of the article is complex heritage-related scene. How does the author define "complex"? Are there any quantitative standards?

Authors: We consider a complex scene to be one whose geometrical characteristics, location and accessibility make a simple data acquisition difficult or impossible (e.g. narrow spaces with little distance between sensor and object). This complexity causes the need to use a large amount of images and/or TLS stations to cover them completely. This is now included in the text. Thank you.

Reviewer 1: 4、The article selects four scenes for experimentation. What is the basis for selecting these four scenes? What are their respective characteristics, and what are the difficulties in conducting photogrammetric measurements? Do they have representativeness?

Authors: Although all scenes show similar characteristics (narrow spaces and difficulties to access and mobility), we have selected these scenes considering some geometrical aspects, trying to include some specific structures such as vertical shafts, corridors and the presence of different chambers separated by narrow accesses.

Reviewer 1: 5、Table 2 lacks an introduction to the column headers.

Authors: We have included these headers in the previous paragraph taking into account your suggestion. Thank you.

Reviewer 1: 6、The description of the experimental results is not accurate enough. In lines 418-419 of the article, it is mentioned that "in all scenes, the maximum values are about 0.03m." However, as shown in Figure 12, only the last two scenes are around 0.03m.

Authors: Changed. Thank you.

Reviewer 1: Comments on the Quality of English Language

The quality of English needs to be improved. Please revise the article throughout.

Authors: The manuscript has been reviewed by a native English speaker. Thank you.

Reviewer 2 Report

Comments and Suggestions for Authors

Although the use of spherical panoramas for photogrammetric processing has been considered many times, this work is of some interest.  In particular, it has been experimentally proven that panoramas created using depth maps show significantly better results than panoramas created without them. The accuracy of photogrammetric processing of such panoramas is close to the accuracy of processing raw fish-eye images. The experiment was carried out correctly: in all cases, the same initial data were used for different processing approaches and a unified control system.
Of particular interest in the work is the demonstration of an approach to scaling a photogrammetric scene based on the distance between multi-camera sensors.

But, in my opinion, some adjustments should be made to the article:

1) The article constantly mentions that the main advantage of panoramic images is the reduction in processing time. It is very important to specifically indicate the time costs for two processing approaches - using FEI images and using SI-DMS images. It is also important to indicate the time spent on obtaining SI-DMS, if it is commensurate with the photogrammetric process. Provide two total times for different approaches to clearly demonstrate how much time we save by sacrificing a little accuracy.

2) In Figure 13, only part d) provides useful information. The rest of the drawing is useless and should be removed. As well as a detailed description of manual model alignment (lines 431-434). Simply mentioning that the model required manual alignment to the CRS is enough.

Author Response

Authors: Thank you for your suggestions and comments. We have rewritten the paper following your suggestions.

Reviewer 2: But, in my opinion, some adjustments should be made to the article:

1) The article constantly mentions that the main advantage of panoramic images is the reduction in processing time. It is very important to specifically indicate the time costs for two processing approaches - using FEI images and using SI-DMS images. It is also important to indicate the time spent on obtaining SI-DMS, if it is commensurate with the photogrammetric process. Provide two total times for different approaches to clearly demonstrate how much time we save by sacrificing a little accuracy.

Authors: We have included a new section (3.3) on processing times in order to analyze the aspect you suggested. Thank you.

Reviewer 2: 2) In Figure 13, only part d) provides useful information. The rest of the drawing is useless and should be removed. As well as a detailed description of manual model alignment (lines 431-434). Simply mentioning that the model required manual alignment to the CRS is enough.

Authors: We agree with you that this explanation can be simplified. However, we think that the inclusion of these figures may help non-specialist readers to understand the manual alignment implemented in this case. Thank you.

Reviewer 3 Report

Comments and Suggestions for Authors

Dear authors, thank you very much for this fascinating work, as use of such cameras is very interesting for confined spaces. Methodology and execution are well thought out and all questions that could rise from readers are answered in your manuscript. There are some minor details, which you may wish to take into account. 

L82-84. The stitching ability of the camera using depth maps is substantial. Given that all relevant information is available (c @ paragrpah 2.1, distance among neighbooring camera lens @ table 3), it would be benefitial if you could add a table with expected depth accuracy at i.e, 1,2,3,5,10 m. Depth accuracy may be estimated as H*H*σpx/c/B.

L173-181. If understood correctly, you imply that the sinply stiching process of the camera (using SIFT, SURF, etc) is less accurate than depth map sticthing, because of illumination variation, parallax, and depth variations, with the later two being correlated. Illumination is exatcly the same to 6 raw fisheye images, from which depth maps are also derived. Hence, illumination cannot be a reason for differentiation. The strong geometric distortions on the edge of each fisheye image, are the main reason for failure for SIFT-like algorithms. Lens distortions provide a different view of each point and surroundings in neighbooring images (where the point might be in the center of one image and on the edge on the next one), hence different descriptors. Your comment on this aspect would be benefitial to readears.

L 286. Why did you prefer Faro (TLS) data capturing for camera calibration, instead of a more traditional approach, such as a refrectroless total station? It is a bit unconventional, so please add a comment

par 3.1.1. and table 3. During the intrinsic camera parameters estimation, (camera calibration), did you also estimated the relative positions and rotations among the cameras? On Table 3 you only provide information about bases, but you must have recovered all 6 parameters of relative (and scale) orientation.

L375-377. Please specify, whether you used this information assuming rigid rig, for the FEI processing. The other option would be to treat rig's parameters as initial and re-estimate in each block, assuming that the rig is not stable, i.e. Overall, please comment on rig's stability/rigitity.

Author Response

Authors: Thank you for your suggestions and comments. We have rewritten the paper following your suggestions.

Reviewer 3: L82-84. The stitching ability of the camera using depth maps is substantial. Given that all relevant information is available (c @ paragrpah 2.1, distance among neighbooring camera lens @ table 3), it would be benefitial if you could add a table with expected depth accuracy at i.e, 1,2,3,5,10 m. Depth accuracy may be estimated as H*H*σpx/c/B.

Authors: We agree with you that the analysis of the error of the depth maps could be of interest in the case of using them directly as result. However, we only analyzed the quality of the stitching procedure but without analyzing the source of its error. This aspect could be studied in future. Thank you for your suggestion.

Reviewer 3: L173-181. If understood correctly, you imply that the sinply stiching process of the camera (using SIFT, SURF, etc) is less accurate than depth map sticthing, because of illumination variation, parallax, and depth variations, with the later two being correlated. Illumination is exatcly the same to 6 raw fisheye images, from which depth maps are also derived. Hence, illumination cannot be a reason for differentiation. The strong geometric distortions on the edge of each fisheye image, are the main reason for failure for SIFT-like algorithms. Lens distortions provide a different view of each point and surroundings in neighbooring images (where the point might be in the center of one image and on the edge on the next one), hence different descriptors. Your comment on this aspect would be benefitial to readears.

Authors: We agree with you. As you suggested, we have removed the illumination variation because it affects equally the three types of stitching. In the same way, we consider that the geometric distortions also affect them equally because the application of SIFT-like algorithms is prior to the stitching procedure. However, the only way to avoid errors caused by parallax or depth variation is to use depth maps. Thank you.

Reviewer 3: L 286. Why did you prefer Faro (TLS) data capturing for camera calibration, instead of a more traditional approach, such as a refrectroless total station? It is a bit unconventional, so please add a comment

Authors: We selected this technique based on two main reasons. First, the positional accuracy of points obtained with TLS is similar over this range of distances. Secondly, because of the fast TLS acquisition and the use of a single station to capture points distributed throughout the scene (including ceiling and floor) in contrast to the need for several stations in case of using a total station (adding errors).

Reviewer 3: par 3.1.1. and table 3. During the intrinsic camera parameters estimation, (camera calibration), did you also estimated the relative positions and rotations among the cameras? On Table 3 you only provide information about bases, but you must have recovered all 6 parameters of relative (and scale) orientation.

Authors: Yes, we have obtained the relative positions and rotations of all fisheye sensors. However, they are not included on Table 3 because they are not used as a constraint for orientation in Metashape software.

Reviewer 3: L375-377. Please specify, whether you used this information assuming rigid rig, for the FEI processing. The other option would be to treat rig's parameters as initial and re-estimate in each block, assuming that the rig is not stable, i.e. Overall, please comment on rig's stability/rigitity.

Authors: In this paragraph we only present the possibility of using FEI without GCPs. We need a complete calibration to achieve it. In section 3.2.2 we describe this approach using intrinsic and extrinsic parameters. The intrinsic parameters (Table 2) are readjusted during the procedure while the extrinsic parameters (Table 3) are used as constrains and are not recalculated, although they consider their quality (given by the STD in Table 3). To readjust these parameters we will need GCPs, which is the process we want to avoid. In this sense, we have tested several configurations and the best results are those shown in the article. This has now been included in text. Thank you.

Reviewer 4 Report

Comments and Suggestions for Authors

The paper is interesting and well structured

Comments on the Quality of English Language

English language level is appropriate

Author Response

Authors: Thank you for your suggestions and comments.

Round 2

Reviewer 1 Report

Comments and Suggestions for Authors

In response to the revision suggestions, the authors have made modifications and explanations. The paper has been improved and enhanced to some extent. However, there are still the following issues:

The description of the experimental results in line 429 is still not accurate enough. The maximum values of QH33SP and QH35P are approximately 0.03m, while QH23 is between 0.015 and 0.02. QH33SB is about 0.01. When describing experimental results, it is important to be as precise as possible, rather than making vague generalizations.

Comments on the Quality of English Language

The quality of English could be further improved.

Author Response

Authors: Thank you for your suggestions and comments. We have rewritten the paper following your suggestions.

Reviewer 1: In response to the revision suggestions, the authors have made modifications and explanations. The paper has been improved and enhanced to some extent. However, there are still the following issues:

The description of the experimental results in line 429 is still not accurate enough. The maximum values of QH33SP and QH35P are approximately 0.03m, while QH23 is between 0.015 and 0.02. QH33SB is about 0.01. When describing experimental results, it is important to be as precise as possible, rather than making vague generalizations.

Authors: This sentence has been changed following your suggestion. Thank you.

Reviewer 1: Comments on the Quality of English Language

The quality of English could be further improved.

Authors: The manuscript has been reviewed by a native English speaker. Thank you.